# Community perceptions and experiences on caring for the premature babies in Arba Minch health and demographic surveillance site, southern Ethiopia: Interpretive Husserlian phenomenological study

**Shitaye Shibiru**[1]*, **Gesila Endashaw**[1], **Mekidim Kassa**[2], **Gistane Ayele**[2], **Agegnehu Bante**[1], **Abera Mersha**[1]

1 School of Nursing, College of Medicine and Health Sciences, Arba Minch University, Arba Minch, Ethiopia,
2 School of Public Health, College of Medicine and Health Sciences, Arba Minch University, Arba Minch, Ethiopia

* shitayeshibiru@yahoo.com

## Abstract

### Introduction

Premature birth is the leading cause of under-5 child mortality, accounting for 18%. More attention is needed for premature babies. Myths, misconceptions, and negative attitudes stigmatize premature births and slow prevention and care efforts. In Ethiopia, studies have been conducted on premature birth and its risk factors. However, there is a lack of evidence regarding the perceived causes of premature birth, caring aspects, and community challenges. This qualitative study aims to address these research gaps.

### Methods

This interpretive Husserlian phenomenological study was conducted from January 1–30, 2022. Purposive sampling was used to recruit 32 participants for focus group discussions and 10 participants for in-depth interviews. Participants included women, grandmothers, grandfathers, men, traditional birth attendants, and traditional healers. Interview and focus group data were analyzed using NVivo 12 Plus software and a thematic content analysis approach.

### Results

In this study, the participants recognized premature babies by physical features such as transparent and bloody bodies, small and weak bodies, a limited range of motion, and bizarre behaviors. They perceived the causes of premature birth to be being young, carrying heavily loaded materials, accidents, illnesses, sin, social influence, and witchcraft. Participants provide warmth to premature babies by wearing cotton wool, making skin-to-skin contact, exposing to sunlight, and wrapping them in clothes. They also feed them boiled

**Data Availability Statement:** All relevant data are within the paper and its Supporting Information files.

**Funding:** b/c Arba Minch University provided a minimal fund by the stated grant number but had no role in study design, data collection and analysis, decision to publish, or preparation of the manuscript.

**Competing interests:** The authors have declared that no competing interests exist.

alcohol, *muk*, and formula, as well as fresh cow milk and butter. They frequently bathe the babies, wash and change their clothes, limit visits, and provide physical protection. The main challenges that the women faced were difficulty feeding and bathing the babies, limited social participation, psychosocial and economic impact, spirituality, and husband negligence.

## Conclusions

The community has a gap in providing care for premature babies, and women with premature babies face many challenges. Therefore, we need to raise awareness of accurate information about the causes and care of premature babies, and we need to support women who have premature babies.

## Introduction

Premature implies babies born alive before 37 weeks of gestation. Based on gestational age, premature birth (PB) is sub-categorized as extremely premature (less than 28 weeks), very premature (28 to 32 weeks), and moderate to late premature (32 to 37 weeks) [1]. Premature babies are prone to severe illness or death during the neonatal period. Even those who survive are at increased risk of lifelong disability and poor quality of life if appropriate care is not provided [2]. There are simple solutions to reduce deaths among premature babies at home in the lowest income settings, such as early and exclusive breastfeeding, handwashing, chlorhexidine application, and skin-to-skin care. Premature babies need extra warmth and support for feeding than others [3, 4].

Worldwide, 15 million babies are born prematurely annually, indicating a global premature birth rate of about 11%. Premature birth complications are the leading cause of under-5 child mortality, accounting for 18%. Besides, 35% of neonatal mortality is also due to prematurity or premature complications globally [1, 5, 6]. The burden of premature birth is high in low- and middle-income countries, especially those in Southeast Asia and sub-Saharan Africa [5]. More than 60% of premature births occur in Africa and South Asia. Within countries, poorer families are at higher risk [1]. Pocket studies in Ethiopia indicated that the prevalence of PB ranged from 4.4% to 16.5% [7–11]. The 3/4th of these deaths could prevent with current and cost-effective interventions [1].

More attention is needed for premature babies in low and middle-income countries. Myths, misconceptions, and negative attitudes make premature births invisible and slow prevention and care efforts [12, 13]. In addition, most community members rely on local newborn illness diagnoses and traditional medications. This has a negative impact on people's willingness to seek medical care for their babies [14].

As shown in qualitative studies from Ghana and Malawi, the perceived causes for premature were categorized into maternal and general social factors. The commonly stated maternal factors were teenage and pregnancies in advanced maternal age, history of abortions and premature birth, prolonged use of family planning method, extramarital sex, sexual impurity, heredity, maternal illness during pregnancy, imbalanced diet, overworking during pregnancy, and husband beating. The general social factors, such as witchcraft and the use of local medicine during pregnancy, were also identified as causes of premature birth [15–17]. One study also suggested that almost all participants recognized an etiology conceptualization and disease

framework for premature birth and distinguished premature birth from miscarriage and macerated stillbirth [1].

Components of care for premature newborns include keeping them warm by using cloth, plastic bottles and hot water bags, squeezing breast milk, and maintaining good hygiene [16]. Other studies also recommend that all premature newborns should be kept warm [3, 18].

The most common challenges to caring for premature newborns were lack of knowledge on how to provide care, poverty, and the high time burden of care, which led to neglect of household, farming, and business duties. Women were mainly responsible for caring for premature newborns [16]. A qualitative study conducted in Canada identified mothers who faced difficulty with breastfeeding, failing to recognize infant feeding distress and disorganized behavior, and the parental stress caused by the multiple feeding issues [19].

In Ethiopia, studies have assessed the prevalence, determinants, and causes of death for premature babies in health facilities and communities [7–11, 18]. However, these studies do not explore how the community perceives the cause, recognition, and care of premature babies, or the challenges they face. Therefore, this qualitative study aimed to fill these research gaps in the study setting.

## Materials and methods

### Qualitative approach, research paradigm, and context

This interpretive Husserlian phenomenological study was conducted in Arba Minch zuria and Gacho Baba district, Arba Minch-Health and Demographic Surveillance Site (AM-HDSS), southern Ethiopia, from January 1 to 30, 2022. Arba Minch-Health and Demographic Surveillance Site was established in 2009 in collaboration between Arba Minch University and the Ethiopian Public Health Association with the support of the Centers for Disease Control and Prevention (CDC) Ethiopia to track demographic changes. The surveillance site includes nine kebeles, selected from the 29 kebeles located in Arba Minch zuria and Gacho Baba districts, Gamo Zone, southern Ethiopia. Arba Minch is an administrative town in the Gamo Zone, located 505 km south of Addis Ababa and 275 km southwest of Hawassa. Based on the 2007 Census conducted by the Central Statistical Agency (CSA), these districts have a total population of 164,529, of whom 82,199 are men and 82,330 are women. According to the HDSS report, there is a total population of 74,157 in the surveillance site.

### Sampling strategy

A purposive sampling method was employed to recruit focus group discussants and key informants. Women with premature babies in less than one year, grandmothers and grandfathers who gave care for premature babies, and men with premature babies in the study setting were selected for focus group discussions (FGD). Traditional birth attendants (TBAs), traditional healers, women, and men were key informants for the in-depth interview (IDI). The number of FGDs and IDIs to meet the objective of this study was determined by idea saturation (Table 1).

### Data collection methods

This study used IDIs and FGDs to collect data. An interview guide was used to collect qualitative data. The interview guide was developed based on the expert opinion, knowledge, and skill of investigators and existing works of literature. The interview guide covered socio-demographic characteristics of study participants, questions on perceived causes of premature birth, how the community recognizes or identifies premature babies, care for premature babies, and

**Table 1. Participants involved to explore community perceptions and experiences on caring for premature new-borns in Arba Minch-health and demographic surveillance site, southern Ethiopia, 2022.**

| Data collection type | Number of participants (n = 42) |
|---|---|
| In-depth interview of the mother with a premature baby | 4 |
| In-depth interview of the father of a premature baby | 2 |
| In-depth interview of the TBA | 2 |
| In-depth interview of the traditional healer | 2 |
| Focus group discussion with the mothers of premature babies | 11 |
| Focus group discussion with the fathers of premature babies | 8 |
| Focus group discussion with the grandmothers | 7 |
| Focus group discussion with the grandfathers | 6 |

challenges in caring for premature babies. This study involved two moderators and two observers for FGDs, and two interviewers for IDIs. Overall, two supervisors were recruited to supervise all FGDs and IDIs. All the moderators, observers, interviewers, and supervisors had at least a master's degree in health-related discipline and previous experience in qualitative data collection. After completing a preliminary session of data collection, including training moderators, observers, interviewers, and supervisors, and pre-testing the interview guide, study participants were traced and advertised with the help of local leaders to recruit them for the study. Then, consent was obtained from those who were willing to participate in this study. Participants who were essential to meet the study objectives were purposively selected by the team and recruited to either FGDs or IDIs. Initially, either the observer or interviewer documented the background information of each participant. The interviews and discussions were held in the participant's homes and open spaces in the village, which was comfortable for the participants. Each FGD and IDI session was recorded using an audio tape recorder. The moderator briefly introduced the team members, explained the study aim, and gave participants the opportunity to ask clarifying questions before initiating the discussion. The discussion was held until idea saturation was reached, or for a maximum of one hour, whichever came first. The moderators controlled the entire discussion and the observers took key notes and recorded all information from the FGDs on the spot. In-depth interviews were held only by the interviewer.

## Trustworthiness, researcher characteristics and reflexivity

Trained and experienced moderators, observers, and interviewers conducted the data collection. The data collectors communicated and discussed daily about any challenges they faced during the data collection period. A well-established, expert-commented, and reviewed interview guide was used to collect the data. Participants with relevant expertise and experience were selected to ensure the trustworthiness of the data. The investigator maintained a neutral view and probed participants to elicit more detailed information. Data credibility was achieved by involving key informants with different social statuses to capture a variety of perspectives and by re-checking original audios and written transcripts to develop themes. NVivo 12 Plus software was used to maintain connections between participant descriptions and the researcher's interpretations, thereby achieving conformability. The results were presented alongside direct quotes from participants. The transcripts, translations and demographic summary of participants were maintained, and keynotes were taken during and after the interview (during analysis) to achieve the dependability of standards [20].

## Data analysis

The collected FGD and IDI audio recordings that were transcribed verbatim in the local language were independently listened to multiple times by the two investigators until they were familiar with the participants' information. Then, the two investigators translated the qualitative data into English transcripts, compared them, and resolved any discrepancies. The investigators read and reread the transcripts to familiarize themselves with the participants' ideas and to develop memos and codes line by line against the themes and subthemes of the framework. They refined and compared the codes for newly emerged themes and subthemes, and discussed their findings daily until they reached agreement on the inconsistencies, new ideas, emerged themes, and subthemes. Data collection continued until idea saturation was reached. Data were analyzed inductively using a thematic content analysis approach with the qualitative data analysis software NVivo 12 Plus [20].

## Ethics approval and consent to participate

This study followed the principles of research ethics adopted by the 64[th] WMA General Assembly, Fortaleza, Brazil, October 2013 [21] and amendments including approval by Arba Minch University, College of Medicine and Health Sciences, Institutional Research Ethics Review Board (IRB). The reference number of the letter was IRB/1047/21. Written and signed voluntary informed consent was obtained from all the discussants and key informants. The anonymity was kept via the use of codes. The study participants also informed that the data (audio records, keynotes, and even transcribed verbatim and transcripts) obtained from them were not accessed by the third party and were kept in utmost confidentiality.

## Results

In this qualitative study, 42 participants, including 32 FGD discussants and 10 key informants, were involved. The median age of participants were 42 years old. Out of the participants, 25 (59.5%) were female, and 16 (38.1%) were not educated, and 3 (7.1%), 12 (28.6%), and 11 (26.2%) were read and write, primary, secondary, and above respectively. In general, seven main themes and twenty-six sub-themes emerged (S1 Table).

## General concept

**Definition for premature babies.** The majority of FGD discussants and IDI key informants defined premature babies as babies born before completing the ninth month. Participants reported that if the mother delivers before the seventh month or at the seventh month and the baby is not alive, it is called abortion *"bosha"*. The majority stated that babies delivered in the eighth month may not survive.

*". . .premature baby means a baby delivered in the seventh month. We know premature birth, as we count the dates starting from the date of cessation of menses and if the baby delivered before the ninth month. . ."* **(Key informant, TBA).**

*". . .if the mother gives birth before the seventh month it is called "bosha" or abortion, if at seventh month and the baby is alive we called that premature baby. If the baby is delivered in the eighth month, the baby cannot survive. After giving birth, I observed my baby, and I told to my mother why we took this premature baby to our home and does this baby will grow. My mother also refused to take my baby when we were discharged from the hospital. We also informed the Doctor to throw the baby in the toilet, but the Doctor counseled us, and we decided to take it home."* **(FGD Discussant, Women).**

**The difference with low birth weight babies.** Almost all participants differentiated premature newborns from low-birth-weight babies. They stated that a low-birth-weight baby is only small or weak, but healthy and delivered after completing the ninth month of pregnancy. Even though premature babies are small and weak, they may be unable to suck and breastfeed. A few participants were unable to differentiate premature babies from low-birth-weight babies.

*"The difference between premature and low birth weight babies is that the low birth weight is small and weak but healthy and delivered after completing the ninth month of pregnancy. Nevertheless, the premature baby is also small and unable to suck the breast, as it is immature."* **(FGD Discussant, Men).**

*"As to me, we cannot differentiate premature newborns from low birth weight baby rather than the health professionals."* **(Key informant, Women).**

## Perceived causes for premature birth

**Unknown cause.** A few participants said that they did not know the exact causes of premature birth, but they shared some community perceptions of the causes.

*". . .I have a son delivered in the seventh month, but now the baby grow. Nevertheless, I do not know the exact cause for premature delivery."* **(FGD Discussant, Grandfather).**

*". . .the community has different perceptions on the causes of premature birth, but we do not know the exact causes for premature delivery."* **(FGD Discussant, Men).**

**Maternal factors.** In this study, most discussants and key informants stated that premature birth can occur if the mother is exposed to different stressful situations. The most common causes of premature delivery cited by participants were being young, carrying heavily loaded materials, accidents, family conflict (including being beaten by a husband or others), and severe illness during pregnancy.

*"As I am very young and I do not know the pregnancy. During that time gush of fluid passed, and I got to the hospital. After the assessment, the Doctor decided to do an operation after informed for me as the amniotic fluid drained completely. So, I perceived that the cause may be related to early pregnancy."* **(FGD Discussant, Women).**

*". . .the causes for premature delivery are fall accidents and carrying heavily loaded material, as a pregnant mother cannot be allowed to grind coffee in a grinder or mortar. The government also not allow pregnant women to do heavy activities, as she did those activities it may cause premature delivery."* **(FGD Discussant, Women).**

*"As an experience, foreigners give special care if women become pregnant; they provide flowers, show feeling of love and give time to refresh their minds. Nevertheless, this is not apply in our community. When women become married and pregnant, they face different challenges such as the husband beating her, others my negligent and this may result in abortion and premature birth."* **(FGD Discussant, Women).**

*"The other cause for premature delivery is that, if the mother has malaria and cannot treated soon, the baby may be delivered premature or "shochetidi kees". After getting sick, if the mother is delayed to get health care, the baby is harmed."* **(FGD Discussant, Women).**

*"My baby was delivered at the seventh month, and during the seventh month of pregnancy, hypertension was diagnosed. Before that, I had anemia and this changed to hypertension. Due to that case, my fetus was unable to get healthy blood. After that, my baby delivered at the seventh month. . ."* **(FGD Discussant, Women).**

**Socio-cultural and spiritual factors.** The community also perceived other causes of premature birth related to socio-cultural issues and spirituality. These causes are believed to be deep-rooted in the community and have existed for a long time as cultural and customary beliefs, such as *"mich"*, *"evil eye"*, and *"ergiman"*. Some other participants also stated spiritually related causes, such as the will of God, *"sin"*, *"gome"* or *"lanche"*, witchcraft, and religious mentalities. There is also social pressure on pregnant mothers to do the same amount of strenuous activity as non-pregnant women.

*". . .my wife was baking injera for others, and she informed that "mich" was one of the causes for premature birth for the first baby and this also true for the second baby even if she stopped the work before six months."* **(Key informant, Men).**

*"The main cause for premature birth is "sin" or "gome" or "lanche" or if someone is not in God way or out of the will of God, and the activity or the circumstance disappointed the family member (father or mother, etc.) or someone else."* **(FGD Discussant, Grandmother).**

*"Our community is also a perception problem if a pregnant mother avoid or is not willing to carry heavily loaded material and other activities, the community rumor that her pregnancy is unique, why she did not do that. Due to that case, the pregnant mother is enforced to do those activities and harm themselves and the concepts. Our community also perceived that "evil eye" can cause premature birth."* **(FGD Discussant, Women).**

*"Most people in our community perceived that the cause for premature birth is the will of God, "witchcraft" or "ergiman"."* **(Key informant, Men).**

### Recognizing premature babies

**Physical features.** The bodies of premature babies are immature, with soft and transparent skin that allows blood vessels and bones to be easily seen. They are also very weak and small.

*"I differentiate my baby as premature as it is very-weak, the skin is very soft, and the body is like blood as it is not mature and the body is transparent. We easily observe the bone and blood vessels of the baby."* **(FGD Discussant, Men).**

*"My baby was delivered in the seventh month, and it looks like "a rat that drown in water". If the baby is delivered in the ninth month the weight is 4 kg, but mine was a 1.8 kg female baby, and the Doctor gave a baby and informed this baby may or may not survive. When they put in the incubator, and the baby becomes unstable. After that, they assessed the health condition and the baby was well."* **(FGD Discussant, Women).**

*". . .the baby looks like a little "chicken" . . . Until now, if we took the baby to health care institution, they told us the baby has malnutrition, and they asked "Does the mother not available for the baby?", and "Does the mother not feed the baby". After that, we told to them that the mother died, and the health professional become anger."* **(FGD Discussant, Grandfather).**

**Limited range of motion.** A few key informants and discussants stated that premature babies have a limited range of motion because their muscles and nerves are not fully developed. Premature babies cannot move their arms and legs freely in the incubator, and their bodies are limp and unresponsive.

*". . .as premature babies are weak, immature, and the body is semis like a paralyzed person . . ."* **(Key informant, Women).**

**Bizarre behaviors.** Premature babies show some unusual characteristics compared to term babies. They are unstable and continuously cry for an unknown reason. When a breast nipple is placed in the baby's mouth, the baby does not show any signs of suckling.

*"Firstly, the baby continuously cries in the seventh month after delivery, and I do not know why the baby continuously cries. Nevertheless, the other individual has told me that the baby cries to indicate a need for a pacifier. They ordered me to provide a pacifier for the baby like a mother breastfeeding rather than wrapping by cloth."* **(FGD Discussant, Grandmother).**

*". . .the premature babies were unstable, continuously crying/shouting, and unable to suck due to discomfort in the abdomen, and I provide herbal medicine for the "evil eye". I also massage the abdomen of the baby for "bua" or hernia."* **(Key informant, Traditional Healer).**

## Community caring practices

**Warmth.** The community provides warmth for premature babies in a variety of ways. After discharge from the healthcare institution, premature babies receive special attention in the community. They are immature and cannot withstand the external weather conditions as well as mature babies. They are dressed in cotton or cotton-wool clothes until they are several months old, skin-to-skin contact (kangaroo mother care or kangaroo father care), sunlight exposure, swaddling, and wrapping with cloth.

*"The premature babies must put in cotton until eight or nine months to give warm and avoid cold."* **(FGD Discussant, Grandfather).**

*". . .I sited in a little chair, put the baby in my chest, and wore a coat to give warmth, and I stayed for 1–4 hours like a monkey. After getting warm for four-hour and the baby started a movement for searching breast to feed."* **(FGD Discussant, Men).**

*"I put the baby in the chest and hold it with a wrapped multiple cloths to make it hot until six months. Until six months, my baby does not go to sick, and the health status was well."* **(FGD Discussant, Women).**

*". . .a premature baby exposed to sunlight after 15 days, as a mother warmth is not enough, but after 30 days for a term baby. Sunlight exposure is important for the baby to gain weight, and the baby exposed the sunlight every other day."* **(FGD Discussant, Grandmother).**

*"My baby discharged early from the hospital as the weight is more than 2kg. So, the care given at home as the skin was very-weak and small. We prevent a baby from cold and moving air by swaddling and wrapping using a cloth made from cotton. We also carefully wash the body as the skin cannot strengthen that much, do not expose it for a long time, and avoid washing while setting in the direction of airflow."* **(Key informant, Men).**

**Feeding.** In the community, fresh cow milk and butter, formula, *"muk"* (a cereal powder-based food), and mother breastmilk were the most common feeding for premature babies. If the baby cannot suck the breast, the baby is fed using a pacifier. Some people may mix fresh cow's milk with boiled alcohol to kill germs.

*"My baby breastfeeds well, but it continuously vomits. Due to that case, I alternatively provide artificially prepared milk or formula "anchor" that I bought from supermarket. . ."* **(FGD Discussant, Women).**

*"The care given for premature babies is giving breast milk and fresh cow milk, and to prevent the child from germs as the baby consumes cow milk, we add boiled alcohol to the fresh milk given by bottle. As fresh cow milk is important to strengthen the body of premature babies and to grow faster. Health professionals recommended supplementary food for babies after six months. But, for premature babies, a "muk" made from barley cereal is also given to the baby in addition to fresh cow mild."* **(Key informant, Women).**

*"I provided fresh butter at six months, and something like a black worm structure come-out from the anus, and we discussed with my mom that this was cause that the baby becomes to be very small. Because there was the woman who delivered at the seventh month with me, and her baby looks like a ninth-month baby."* **(FGD Discussant, Women).**

**Hygiene.** Premature babies need frequent bathing (2–3 times per day) and clothes changes to prevent infection and other complications, as their immune systems are immature. However, some study participants raised concerns that frequent bathing could expose premature babies to cold. To reduce the risk of infection, it is important to clean equipment, wash breasts before feeding, and use wipes rather than cloth or other materials after passing urine or stool.

*"The body of a premature baby must wash continuously (two to three times per day) to make it clean and comfortable."* **(FGD Discussant, Men).**

*"We bathed the baby every week to prevent cold while immersing frequently into the water as the body was immature. Again, after passing stool, we used a wipe to clean as washing with water or using other cloths are not much recommend for premature babies."* **(Key informant, Men).**

*"I kept the hygiene by washing his body and changing the clean cloths frequently, and I washed my breast before giving it to him to suck, and all the equipment used for the baby must be clean."* **(Key informant, Women).**

**Limit visiting.**   A few participants also reported that limiting the number of visitors is essential to prevent infection in premature babies. Premature babies are highly susceptible to infection because their immune systems are immature. This also helps to avoid discouraging new parents.

*"We limited the number of persons who visit the baby as informed by health care professionals to prevent infection and tease from other people."* **(FGD Discussant, Women).**

**Physical protection.**   Premature babies must be physically protected because they are very small and weak. The discussants and key informants stated that premature babies are easily harmed because their bodies are immature and soft.

*"I have not put my baby in the bed, as the baby was very-small and even it was difficult to differentiate the leg and the arm, and I am much suspicious as someone my sitting on the baby. As such, it needs critical care (care during breastfeeding and washing the body)."* **(FGD Discussant, Women).**

## Support for the mother with premature baby

**Family support.**   Premature babies need critical follow-up, and their mothers suffer more than others. As such, mothers who have given birth prematurely need support from their families (husbands, mothers-in-law, grandmothers, grandfathers, etc.). This support can include helping them with activities of daily living and participating in social affairs.

*"My husband support and allow me to do different household activities and to go social participations like "lekiso" or rituals, and other related social activities. After having those activities, I returned to the home, and my husband alternatively does those activities as caring for a premature newborn is very challenging."* **(FGD Discussant, Women).**

*"The support from the family is needed for the mother who faced such challenges like the husband, grandmother, grandfather and the mother-in-law, even neighbors must care for the baby and the mother must inform for those supports . . ."* **(Key informant, Women).**

**Community or social support.**   The community must support mothers of premature babies during their care. A premature baby needs a caregiver who is fully present and attentive, and it is important for the mother to not have to leave the baby alone for even a few minutes. Because close follow-up of the baby limits the mother's ability to participate in society, the community should allow these mothers to take a break and provide psychological support.

*". . .the neighbors and relatives may support the mother with a premature baby, and the local people and the kebele leaders understand the situation and give permission for the family until the baby strengthens him/herself."* **(FGD Discussant, Women).**

*"In our case, related to social life, the community understands that it is not mandatory until six months whether the baby is delivered term or prematurely. If the attendances called in "Idir" or "burial ceremony", and other activities, their neighbor informed them that she is "mechat" or mother of the baby."* **(FGD Discussant, Men).**

**Health professionals' support.** All information about premature babies should be discussed during antenatal care, and the community must be aware of this through different mechanisms. Health professionals in the surrounding area also supported the mother with the premature baby by counseling on caring practices, such as feeding, bathing, maintaining warmth, and hygiene. Health professionals must provide information and mothers must be informed to avoid misconceptions in the community.

*"The health professionals in the health center gave counselling to the pregnant mother during antenatal care to avoid carrying heavily loaded materials, to have a balanced diet, and to keep herself from any situations that harm the mother as well as the fetus."* **(FGD Discussant, Men).**

*"The health extension workers also informed all the pregnant mothers and the husbands to avoid heavy loaded materials or equipment during pregnancy as it may harm the conceptus."* **(Key informant, Women).**

*"My wife had a follow-up in the health center, and she got counseling service from health care providers regarding how to care for premature babies are given." (***Key informant, Women).**

**Government support.** The government also plays a role in supporting mothers with premature babies by designing different strategies to raise community awareness and improve access to nearby health facilities and services.

*"As previously, the government provide "aja" for those mothers who are from low-income family and for premature babies also. The government also avail health extension workers at kebele level, and those provide different health-related information."* **(FGD Discussant, Men).**

## Challenges the women faced

**Difficult to feed.** Almost all of the study participants stated that it is very challenging to feed premature babies. Premature babies are cannot suck and fully breastfeed.

*". . .that premature babies do not suck, and it is challenging to feed them."* **(FGD Discussant, Women).**

*"I faced a great challenge in caring for my baby as it does not fully breastfeed and as most people said such like the baby does not grow, and it may die or pathway."* **(Key informant, Women).**

**Difficult to bath.** Bathing premature babies is challenging because their bodies are tiny and fragile. Some mothers panic when they see their premature babies for the first time. The skin of premature babies is easily bruised and very delicate, so it is important to be gentle when bathing them.

*"It is difficult to bathe the baby as the body is small and weak. Even, some mothers do not allow to unclothe and are frightened while seeing the body."* **(Key informant, TBA).**

**Limit social participation.** Discussants and key informants explained that having a premature baby can restrict social participation. The mother may not feel comfortable leaving the baby at home with others or participating in community activities or social life.

*"My wife stayed at home to look for the baby, and she is limited for social issues. Nevertheless, I actively engaged in social activities like "Ikub", "Idir", "Yeho", and "Debo". As such, the neighbors know the situation, and they permit her to keep her baby."* **(Key informant, Men).**

*". . .it is difficult to grow premature babies as it needs close follow-up. It limits social interaction and other activities like going to the market, attending social ceremonies, and cooking food. The mother becomes hungry if someone is not around her as the baby continuously cries, and the mother does not cook food for him even after one month of delivery."* **(FGD Discussant, Grandmother).**

*"I had not attended social graves or rituals, ceremonies, and festivities for long time, and the neighbors, the kebele administrates also understand the situation and they supported me."* **(FGD Discussant, Women).**

**Prone for infection or any disease.** Premature babies have a less developed immune system, so they are more susceptible to infection than full-term babies. Vomiting and diarrhea are common symptoms of infection in premature babies.

*"It is very challenging that the baby faced diarrhea, vomiting, and tonsillitis. The baby is weak until now because he cannot get breast milk."* **(FGD Discussant, Grandmother).**

*". . .after discharging from the hospital, the babies developed wounds throughout the body, and the baby semis like a "pen". We challenged to manage and very stressed. My husband continuously prays to God that the baby becomes well. When we took the baby to the hospital, the health professional always asked why this baby is wasted as you are the mother of baby and you are well enough. Even if, I responded to them the baby was delivered in the seventh month, they cannot accept my response."* **(FGD Discussant, Grandmother).**

**Psychosocial and economic impact.** Delivering a premature baby in the seventh month can cause stress for families, who may find caring for the baby to be challenging and worry that the baby may not survive. Some people in the community believe that premature birth is caused by *"sin"* or *"amilko"*, and that premature babies born in the eighth month will not survive. Caring for a premature baby can also have a negative financial impact on families, due to unexpected costs and the need to reduce working hours.

*"The baby delivered in the seventh month is challenging as the families are in trouble and stressed. The mother has faced a challenge if she gives birth to a premature baby."* **(FGD Discussant, Men).**

*"Growing a premature baby is challenging as psychosocially and economically affects the family. If a premature baby is born in the surrounding and the community perceive that this baby does not survive and always talks about that baby and this creates great stress on family members."* **(FGD Discussant, Women).**

*"We faced a challenge from the community that all are forwarded a comment that it is better if this baby was born in the seventh month as a baby born in the eighth month cannot survive. I believe an eighth-month baby is better than a seventh-month baby because the eighth-month baby stays a month long in the mother's womb. Even, after being discharged from the hospital, most people believed this baby might die after two or three days. Nevertheless, the baby survived until now without any problem."* (**Key informant, Men**).

**Lack of support from husband.**    The other main challenge for the mother of premature babies is the lack of support from her husband.

*"Mothers face the main challenge either during pregnancy or during caring for the child. The husband is simply as the name husband rather than sharing the burden of the wife."* (**FGD Discussant, Grandfather**).

## Compulsory action

**Lack of hospitals in the surrounding.**    Premature babies need immediate access to advanced health services, but these services are often unavailable or inaccessible. This is a serious problem that needs to be addressed immediately. Pregnant mothers suffer greatly when their babies are premature, and the lack of access to advanced health services can make the situation even more difficult.

*"Pregnant mothers suffer more to rich to hospitals as there is no hospital in our surroundings, and mothers must go to the hospital via transport from the local health center if complications occur. We carry the pregnant mother via chair or bed to reach the main road to take to the hospital."* (**FGD Discussant, Men**).

**Lack of support from local health workers.**    As reported by the FGD discussants and key informants for IDI, support from community health workers or health extension workers (HEWs) is needed. They should closely follow premature babies after discharge from the hospital and create awareness of premature birth in the community.

*"After discharge from the hospital, as my brother is Doctor and he wrote all the information about my baby and informed me to give this written paper for HEWs in which the baby needs close follow-up. Nevertheless, they do not give support as my baby is unstable and seriously ill, even though they do not ask me about the situation. Due to that case, I consulted a traditional healer, he ordered traditional herbal medicine and informed me to re-visit again after two days, and he provided the medicine again. Then, the baby becomes stable without going to a healthcare institution. The herbal given to the baby was for "evil eye"."* (**FGD Discussant, Men**).

*". . .said that the health extension workers in our surrounding do not follow the mothers who delivered in seven months. They also do not provide any information related to pre-term delivery rather than routine service like vaccination based on the ordered schedule and as provided in the campaign."* (**FGD Discussant, Women**).

*"Support from health workers is needed. Health professionals create awareness of premature babies through house-to-house visits, how those babies carried, does it grow or not. I feel that*

*my baby lost due to my supplementation of fresh cow milk due to lack of knowledge on feeding premature babies."* (**Key informant, Women**).

**Poor road construction and lack of transportation.**   The other focus area that the participants stated is poor road construction and lack of transportation to hospitals and clinics that provide qualified or advanced services.

*"The main challenge related to this is that it is difficult to rich the pregnant mother to the hospital, as the road is not well structured and we cannot access transportation."* (**FGD Discussant, Men**).

## Discussion

Premature birth complications are a public health issue and the main contributor to neonatal mortality. Developed nations have seen a progressive decrease in neonatal mortality, but underdeveloped and developing nations have not. Despite different interventions, neonatal mortality has raised in Ethiopia. As such, it needs attention that premature birth increases from time to time. There are myths, misconceptions, and negative feelings about caring for premature babies in the community. A few studies were conducted in our country Ethiopia on premature babies. Nevertheless, there is a lack of evidence on perceived causes of premature birth, caring aspects, and challenges in the community. Therefore, this qualitative study aimed to fill those research gaps in the study setting.

In this study, participants defined premature birth as the delivery before the ninth month of pregnancy. The perceived causes for premature birth were related to maternal, socio-cultural, and spiritual factors. The most common ways of recognizing premature babies were physical features, neuromuscular immaturity, and by their unusual characteristics. Almost all of the discussants and key informants reported that thermal protection for the baby was provided by putting in cotton and skin-to-skin contact, feeding cow milk, and *"muk"* in addition to the mother's breast milk. Mothers with premature babies frequently bathe them to keep them clean, limit visitors to prevent infection, and physically protect them. The community, family, health professionals, and the government should support mothers of premature babies. Mothers with premature babies face challenges related to feeding, bathing, and social participation. They also resulting in different costs as premature babies are prone to infection, psychosocial and economic impact, and lack support from their husband. The top prioritized compulsory action that the study participants stated was the lack of hospitals in the surrounding, lack of support from local health workers, poor road construction, and lack of transportation.

In this finding, the majority of the participants stated that premature babies as babies delivered before completing the ninth month of pregnancy. However, a few participants challenged to differentiate premature newborns from low-birth-weight babies. This was also supported by the studies conducted in Malawi [1, 16], Ghana [15], and Uganda [22]. This is likely because most laypersons relate prematurity with size and consider any baby with a very small birth weight to be premature. Therefore, it is important to raise awareness in the community about the difference between premature babies and low-weight babies. This can be done by community health workers.

A study conducted in Malawi reported that the perceived causes of premature birth were categorized as maternal and general social factors. Maternal factors included not eating good-

quality and enough food, doing excess household chores, being beaten by one's husband, frequent illnesses, having a previous abortion, having a family history of premature birth, and early or late childbearing. General social factors included the will of God, witchcraft, and the use of local medicine during pregnancy [16]. Similarly, a study conducted in Ghana speculated that teenage pregnancies, unsafe abortions, weak sperm of men, prolonged use of family planning methods, extramarital sex by the father, and witchcraft were perceived causes of premature birth. The beliefs about the causes of premature births were either mystical or natural phenomena. Beliefs about the mystical/supernatural causes mentioned were witchcraft, ancestral disagreements with the family due to disregard of what they require, powers of "wicked trees" and rocks within the surroundings, and very old animals, such as dogs and pigs, that live in the house [15]. A study from Uganda reported that biomedical-related explanations for the causes of premature births included diseases (syphilis and malaria) or other medical complications. However, many community members associated premature births with causes such as witchcraft from a co-wife, the will of God, and the occurrence of earthquakes (called musisi in the local language) [22]. Adolescents give birth to premature babies more often than adults, as indicated in a study done in Cameroon [23]. This finding also reported that being young, carrying heavily loaded materials, accidents, family conflict, and severe illness during pregnancy were identified as maternal factors during pregnancy that causes premature delivery. On the other hand, "*mich*", "*evil eye*", "*ergiman*", the will of God or "*sin*", "*gome*" or "*lanche*", and witchcraft were the perceived causes for prematureness related to socio-culture and spirituality.

In this study, participants recognized premature babies by their physical features, neuromuscular maturity, and bizarre behaviors. Physical features they stated included soft skin, a transparent body (with blood vessels and bones easily visible), and being very weak and small. Limited range of motion (unable to flex and extend extremities) was related to neuromuscular maturity. The most commonly stated bizarre (unusual) characteristics were instability, continuous crying, and not showing signs of suckling. In line with this, studies conducted in Ghana, Uganda, and Malawi mentioned the following physical features of premature babies at birth: very small and transparent bodies that are soft, unable to suckle and breastfeed, absence of eyelashes, sunken forehead, many wrinkles, breathlessness, and floppy muscles [15, 16, 22].

This study found that the participants provided warmth for the premature babies by putting in cotton wool, skin-to-skin contact (kangaroo mother and kangaroo father care), sunlight exposure, swaddling, and wrapping with cloth. Different works of literature reported that warmth for premature newborns was universal care. It is provided by wrapping cloth, making fire inside the house, closing windows and doors, and keeping the baby inside the house all the time [3, 16, 18]. In contrast, most participants were little known and not practice skin-to-skin care (kangaroo mother care) and some practices were not appropriate that may harm the baby unless controlled, such as lighting lamps and charcoal stoves placed under the babies bed, and hot water jerry cans or plastic bottles put close to the baby [3, 16, 18, 22]. There is also evidence that the mother should provide kangaroo mother care whenever possible, and if the mother is not available, fathers, partners, and other family members can also provide [18, 24].

A study from Malawi found that expressing breast milk (mothers squeezing breast milk into a cup and using a spoon to feed the newborn) is a common practice for premature babies [16]. There is strong evidence that providing breast milk is the standard of care across all countries and the core of many national policies and programs for premature babies [24]. In contrast, this study found that the participants in the community provided fresh cow milk and butter, formula, "*muk*" and mother's breast milk using a pacifier. A few participants may have mixed fresh cow milk with boiled alcohol to kill germs. Similarly, a study from Uganda found that traditional birth attendants (TBAs) advised giving sugar water to premature babies if the

mother perceived that she did not have enough breast milk [22]. This is malpractice, and it can harm premature babies and result in various complications. Therefore, it is best to avoid giving premature babies anything other than breast milk until they are six months old.

Immediate and frequent bathing, changing clothes, cleaning equipment, washing breasts before feeding, and using wipes were the most common practices used by most of the study participants to keep premature babies hygienic. A few also reported bathing the baby after a week of delivery to prevent a cold. On the contrary, a study from Malawi reported that a premature newborn was not bathed until it reached nine months of corrected gestational age. A few kept the house clean by washing the newborn's clothes and sprinkling water on the house to control dust [16]. Another study indicated that the baby was bathed immediately after delivery and cooking oil was used on its body [22].

Almost all of the study participants in Malawi reported that premature newborns were not allowed to see other people until they reached nine months of corrected gestational age [16]. The main reasons they stated were the high susceptibility of premature babies to infection and constant dissuasion from other people. The discussants of this study also raised a similar idea. The mother, family, or anyone who provides care to the premature baby has a responsibility to physically protect or safeguard them from harm or injury even more than term babies. This point, stated by the discussants and key informants of this study, is that premature babies are very small and weak and can be easily harmed because their bodies are immature and soft.

Community health workers advised mothers with premature babies on newborn care practices and promoted malaria prevention and family planning [22]. Similarly, home visits are recommended to support families in caring for their premature or low-birth-weight infants, as this can increase exclusive breastfeeding, immunization visits, and parental-infant attachment, and decrease parental stress and anxiety [24]. This study also reported that health professionals in the surrounding supported the mother with a premature baby by counseling on how the mother handles the baby (feeding, bathing, maintaining warmth, hygiene, etc.). Families of premature or low-birth-weight infants should be given extra support and must involve in routine care. Those supports include education, counseling and discharge preparation by health workers, and peer support [24]. In the same way, the participants of this study also speculated that family, community or society, and government support are needed for mothers with premature babies.

Mothers with premature babies faced challenges while providing care. These challenges included premature babies falling sick often, resulting in poverty, failure to do business and household chores, men starting sexual affairs outside marriage, and mother's lack of knowledge on how to properly care for them [16]. In line with this, study found that mother challenged to feed and bathe due to lack of knowledge, limits social participation, premature babies being prone to infection, psychosocial and economic impact, and women lack support from husbands. The compulsory action reported by the discussants and key informants were the lack of a hospital in the surroundings, lack of support from local health workers, poor road construction, and lack of transportation.

The limitation of this study is that, due to its phenomenological nature, it may be affected by subjectivity and researcher-induced bias to some extent. Therefore, these factors must be considered when interpreting the findings of this study.

This study highlighted a current public health issue and generated evidence that can be used to plan interventions. Policymakers and program planners can use these findings to design appropriate strategies for care practices for premature babies in the community. Deeply rooted practices in the community may put premature babies at further risk of complications. This study identified these practices and provided a roadmap for interventions and research.

## Conclusions

The community has a gap in providing care for premature babies, and women with premature babies face challenges. There are malpractices in the community related to feeding and hygiene keeping for premature babies and the causes of prematurity. The challenges identified include difficulty and lack of knowledge on feeding and bathing premature babies, susceptibility of the babies to infection, psychosocial and economic impact, limiting social participation, and problems related to the husband. These findings are original and contribute to knowledge on this topic. Therefore, attention and awareness should be raised to avoid misconceptions about causes and caring practices, and there is a need to share the burden of women.

## Supporting information

**S1 Checklist.**
(DOCX)

**S1 Dataset. Participant information.**
(XLSX)

**S2 Dataset. Translated qualitative information.**
(DOCX)

**S1 File. English version of interview script.**
(PDF)

**S2 File.**
(JPG)

**S1 Table. Developed main themes and sub-theme.**
(DOCX)

## Acknowledgments

The authors are pleased to express their appreciation to Arba Minch University, College of Medicine and Health Sciences, and Arba Minch-Health and Demographic Surveillance Site Coordination Office for providing financial support for this research project. We would like to extend our deepest appreciation and thanks to our colleagues for their unwavering support throughout the research activities and for providing constructive comments and advice. Finally, our heartfelt thanks also go to the key informants, discussants, data collectors (interviewers, moderators, and observers), supervisors, coordination office workers, and field data collectors who provided baseline information about the study area.

## Author Contributions

**Conceptualization:** Shitaye Shibiru.

**Data curation:** Shitaye Shibiru, Gesila Endashaw, Mekidim Kassa, Gistane Ayele, Agegnehu Bante, Abera Mersha.

**Formal analysis:** Shitaye Shibiru, Abera Mersha.

**Funding acquisition:** Shitaye Shibiru, Abera Mersha.

**Investigation:** Shitaye Shibiru, Abera Mersha.

**Methodology:** Shitaye Shibiru, Abera Mersha.

**Project administration:** Shitaye Shibiru, Abera Mersha.

**Resources:** Shitaye Shibiru, Agegnehu Bante, Abera Mersha.

**Software:** Shitaye Shibiru, Abera Mersha.

**Supervision:** Shitaye Shibiru, Abera Mersha.

**Validation:** Shitaye Shibiru, Gesila Endashaw, Mekidim Kassa, Gistane Ayele, Abera Mersha.

**Visualization:** Shitaye Shibiru, Abera Mersha.

**Writing – original draft:** Shitaye Shibiru, Abera Mersha.

**Writing – review & editing:** Shitaye Shibiru, Gesila Endashaw, Mekidim Kassa, Gistane Ayele, Agegnehu Bante, Abera Mersha.

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
