## [Decision Letter · Decision Letter 0]

10 Oct 2023

PONE-D-23-19601Community perceptions and experiences on caring for the premature babies in Arba Minch Health and Demographic Surveillance Site, southern Ethiopia: Interpretive husserlian phenomenological studyPLOS ONE

Dear Dr. Shibiru,

Thank you for submitting your manuscript to PLOS ONE. After careful consideration, we feel that it has merit but does not fully meet PLOS ONE’s publication criteria as it currently stands. Therefore, we invite you to submit a revised version of the manuscript that addresses the points raised during the review process.

We look forward to receiving your revised manuscript.

Kind regards,

Madhulika Sahoo, Ph.D

Academic Editor

PLOS ONE

Reviewers' comments:

Reviewer's Responses to Questions

**Comments to the Author**

1. Is the manuscript technically sound, and do the data support the conclusions?

Reviewer #1: Yes

Reviewer #2: Yes

2. Has the statistical analysis been performed appropriately and rigorously? 

Reviewer #1: Yes

Reviewer #2: Yes

3. Have the authors made all data underlying the findings in their manuscript fully available?

Reviewer #1: Yes

Reviewer #2: Yes

4. Is the manuscript presented in an intelligible fashion and written in standard English?

Reviewer #1: Yes

Reviewer #2: No

5. Review Comments to the Author

Reviewer #1: The paper is technically sound. The author(s) have used appropriate statistical techniques for analysing the data. The necessary data required has been made available. The manuscript is well organised and written in simple english with illustration and explanations. This is a timely contribution and is ready for publication.

Reviewer #2: This is a well written manuscript highlighting a major issue of community perception of care for preterm babies in Southern Ethiopia. The methodology is sound and as per my knowledge the statistical analysis has been well performed. The findings have been well reflected in the conclusion. The data underlying the findings in the manuscript has been made available including the questionnare in English language and Ethical approval. Regarding the references the author has added credible sources such as WHO and more recent references have been used. In my view this study and gap identified is significant and there is no duplicate study on Community perceptions and experiences on caring for the premature babies in Arba

Minch Health and Demographic Surveillance Site, southern Ethiopia.

After going through this manuscript thoroughly my opinion is that this study is suitable for publication subject to some minor corrections given below:

-Grammatical error should be corrected

- The English language needs to be improved in the text.

- Proofreading is required

6. PLOS authors have the option to publish the peer review history of their article (what does this mean?). If published, this will include your full peer review and any attached files.

Reviewer #1: **Yes: **SANGHMITRA SHEEL ACHARYA

Reviewer #2: **Yes: **Dr Nimra Iqbal Choudhary

---

## [Author Response · Author response to Decision Letter 0]

13 Oct 2023

Dear Editor,

Thank you for your time and constructive feedback on our manuscript. I have responded to each reviewer's comments and suggestions in detail below.

1. Please ensure that your manuscript meets PLOS ONE's style requirements, including those for file naming

Response: I thank you for giving this suggestion to cross-check the journal requirements. I cross-checked the PLOS ONE's style requirements and the manuscript meets the requirements. 

Response: That information directly submitted or inputted to the system rather than included in the manuscript. It was checked and the information included was correct, b/c Arba Minch University provided a minimal fund by the stated grant number but had no role in study design, data collection and analysis, decision to publish, or preparation of the manuscript.

3. In your Data Availability statement, you have not specified where the minimal data set underlying the results described in your manuscript can be found.

Response: That information directly submitted or inputted to the system rather than included in the manuscript. The minimal data set underlying the results was described in the revised manuscript (included as supporting information).

4. Please review your reference list to ensure that it is complete and correct.

Response: I cross-checked and all the references were complete and correctly stated. 

Reviewer #2:

-Grammatical error should be corrected

- The English language needs to be improved in the text.

- Proofreading is required

Response: The grammatical errors were corrected, the English language was improved, and the manuscript was proofread.

Best Regards!

Shitaye Shibiru,

On behalf of co-authors

---

## [Editor Report · Decision Letter 1]

26 Oct 2023

Community perceptions and experiences on caring for the premature babies in Arba Minch Health and Demographic Surveillance Site, southern Ethiopia: Interpretive Husserlian phenomenological study

PONE-D-23-19601R1

Dear Dr. Shibiru,

We’re pleased to inform you that your manuscript has been judged scientifically suitable for publication and will be formally accepted for publication once it meets all outstanding technical requirements.

Kind regards,

Madhulika Sahoo, Ph.D

Academic Editor

PLOS ONE
---

## [Editor Report · Acceptance letter]

3 Nov 2023

PONE-D-23-19601R1 

Community perceptions and experiences on caring for the premature babies in Arba Minch Health and Demographic Surveillance Site, southern Ethiopia: Interpretive Husserlian phenomenological study 

Dear Dr. Shibiru:

I'm pleased to inform you that your manuscript has been deemed suitable for publication in PLOS ONE. Congratulations! Your manuscript is now with our production department. 

Kind regards, 

on behalf of

Dr. Madhulika Sahoo 

Academic Editor

PLOS ONE